# GAN Memory with No Forgetting

Yulai Cong[*]    Miaoyun Zhao[*]    Jianqiao Li    Sijia Wang    Lawrence Carin
Department of Electrical and Computer Engineering
Duke University

## Abstract

As a fundamental issue in lifelong learning, catastrophic forgetting is directly caused by inaccessible historical data; accordingly, if the data (information) were memorized perfectly, no forgetting should be expected. Motivated by that, we propose a GAN memory for lifelong learning, which is capable of remembering a stream of datasets via generative processes, with *no* forgetting. Our GAN memory is based on recognizing that one can modulate the "style" of a GAN model to form perceptually-distant targeted generation. Accordingly, we propose to do sequential style modulations atop a well-behaved base GAN model, to form sequential targeted generative models, while simultaneously benefiting from the transferred base knowledge. The GAN memory – that is motivated by lifelong learning – is therefore itself manifested by a form of lifelong learning, via forward transfer and modulation of information from prior tasks. Experiments demonstrate the superiority of our method over existing approaches and its effectiveness in alleviating catastrophic forgetting for lifelong classification problems. Code is available at `https://github.com/MiaoyunZhao/GANmemory_LifelongLearning`.

## 1 Introduction

Lifelong learning (or continual learning) is a long-standing challenge for machine learning and artificial intelligence systems [76, 28, 73, 11, 14, 60], concerning the ability of a model to continually learn new knowledge without forgetting previously learned experiences. An important issue associated with lifelong learning is the notorious catastrophic forgetting of deep neural networks [48, 36, 87], *i.e.*, training a model with new information severely interferes with previously learned knowledge.

To alleviate catastrophic forgetting, many methods have been proposed, with most focusing on discriminative/classification tasks [36, 65, 95, 55, 94]. Reviewing existing methods, [77] revealed generative replay (or pseudo-rehearsal) [72, 69, 86, 66, 88] is an effective and general strategy for lifelong learning, with this further supported by [40, 78]. That revelation is anticipated, for if the characteristics of previous data are remembered perfectly (*e.g.*, via realistic generative replay), no forgetting should be expected for lifelong learning. Compared with the coreset idea, that saves representative samples of previous data [55, 65, 11], generative replay has advantages in addressing privacy concerns and remembering potentially more complete data information (via the generative process). However, most existing generative replay methods either deliver blurry generated samples [10, 40] or only work well on simple datasets [40, 78, 40] like MNIST; besides, they often don't scale well to practical situations with high resolution [60] or a long sequence [86], sometimes even with negative backward transfer [96, 82]. Therefore, it's challenging to continually learn a well-behaved generative replay model [40], even for moderately complex datasets like CIFAR10.

We seek a realistic generative replay framework to alleviate catastrophic forgetting; going further, we consider developing a realistic generative memory with growing (expressive) power, believed to be a fundamental building block toward general lifelong learning systems. We leverage the popular

---

[*]Equal Contribution. Correspondence to: Miaoyun Zhao <miaoyun9zhao@gmail.com>.

GAN [25] setup as the key component of that generative memory, which we term GAN memory, because (*i*) GANs have shown remarkable power in synthesizing realistic high-dimensional samples [9, 51, 33, 34]; (*ii*) by modeling the generative process of training data, GANs summarize the data statistical information in the model parameters, consequently also protecting privacy (the original data need not be saved); and (*iii*) a GAN often generates realistic samples not observed in training data, delivering a synthetic data augmentation that potentially benefits better performance of downstream tasks [80, 8, 9, 21, 33, 26, 27]. Distinct from existing methods, our GAN memory leverages transfer learning [6, 16, 46, 92, 58] and (image) style transfer [18, 30, 41]. Its key foundation is a discovery that one can leverage the modified variants of style-transfer techniques [64, 98] to modulate a source generator/discriminator into a powerful generator/discriminator for perceptually-distant target domains (see Section 4.1), with a limited amount of style parameters. Exploiting that discovery, our GAN memory sequentially modulates (and also transfers knowledge from) a well-behaved base/source GAN model to realistically remember a sequence of (target) generative processes with *no* forgetting. Note by "well-behaved" we mean the shape of source kernels is well trained (see Section 4.1 for details); empirically, this requirement can be *readily satisfied* if (*i*) the source model is pretrained on a (moderately) large dataset (*e.g.,* CelebA [43]; often a dense dataset is preferred [85]) and (*ii*) it's sufficiently trained and shows relatively high generation quality. Therefore, many pretrained GANs can be "well-behaved",[2] showing great flexibility in selecting the base/source model. Our experiments will show that flexibility roughly means source and target data should have the same data *type* (*e.g.*, images).

Our GAN memory serves as a solution to the fundamental memory issue of general lifelong learning, and its construction also leverages a form of lifelong learning. In practice, the GAN memory can be used, for example, as a realistic generative replay to alleviate catastrophic forgetting for challenging downstream tasks with high-dimensional data and a long (and varying) task sequence. Our contributions are as follows.

- Based on FiLM [64] and AdaFM [98], we develop modified variants, termed mFiLM and mAdaFM, to better adapt/transfer the source fully connected (FC) and convolutional (Conv) layers to target domains, respectively. We demonstrate that mFiLM and mAdaFM can be leveraged to modulate the "style" of a source GAN model (including both the generator and discriminator) to form a generative/discriminative model capable of addressing a perceptually-distant target domain.

- Based on the above discovery, we propose our GAN memory, endowed with growing (expressive) generative power, yet with *no* forgetting of existing capabilities, by leveraging a limited amount of task-specific style parameters. We analyze the roles played by those style parameters and reveal their further compressibility.

- We generalize our GAN memory to its conditional variant, followed by empirically verifying its effectiveness in delivering realistic synthesized samples to alleviate catastrophic forgetting for challenging lifelong classification tasks.

## 2 Related work

**Lifelong learning** Exiting lifelong learning methods can be roughly grouped into three categories, *i.e.,* regularization-based [36, 69, 95, 55, 68], dynamic-model-based [47, 68, 47], and generative-replay-based methods [72, 42, 86, 66, 88, 78]. Among these methods, generative replay is believed an effective and general strategy for lifelong learning problems [66, 88, 40, 78], as discussed above. However, most existing methods of this type often have blurry/distorted generation (for images) or scalability issues [86, 66, 59, 60, 96, 54, 82]. MeRGAN [86] leverages a copy of the current generator to replay previous data information, showing increasingly blurry historical generations with reinforced generation artifacts [96] as training proceeds. CloGAN [66] uses an auxiliary classifier to filter out a portion of distorted replay but may still suffer from the reinforced forgetting, especially in high-dimensional situations. OCDVAE [54] unifies open-set recognition and VAE-based generative replay, whereas showing blurry generations for high-resolution images. Based on a shared generator, DGMw [59] introduces task-specific binary masks to the generator weights, accordingly suffering from scalability issues when the generator is large and/or the task sequence is long. Lifelong GAN [96] employs cycle consistency (via an auxiliary encoder) and knowledge distillation (via copies of

that encoder and the generator) to remember image-conditioned generation; however, it still shows decreased performance on historical tasks. By comparison, our GAN memory delivers realistic synthesis with *no* forgetting and scales well to high-dimensional situations with a long task-sequence, capable of serving as a realistic generative memory for general lifelong learning systems.

**Transfer learning** Being general and effective, transfer learning has attracted increasing attention recently in various research fields, with a focus on discriminative tasks like classification [6, 70, 90, 93, 16, 24, 44, 46, 74, 92] and those in natural language processing [3, 63, 53, 52, 2]. However for generative tasks, only a few efforts have been made [85, 58, 98]. For example, a GAN pretrained on a large-scale source dataset is used in [85] to initialize the GAN model in a target domain, for efficient training or even better performance; alternatively, [58] freezes the source GAN generator only to modulate its hidden-layer statistics to "add" new generation power with $L1$/perceptual loss; observing the general applicability of low-level filters in GAN generator/discriminator, [98] transfers-then-freezes them to facilitate generation with limited data in a target domain. Those methods either only concern synthesis in the target domain (completely forgetting source generation) [85, 98] or deliver blurry target generation [58]. Our GAN memory provides both realistic target generation and no forgetting on source generation.

## 3 Preliminary

We briefly review two building blocks on which our method is constructed: generative adversarial networks (GANs) [25, 33] and style-transfer techniques [64, 98].

**Generative adversarial networks (GANs)** GANs have shown increasing power to synthesize highly realistic observations [32, 45, 51, 9, 33, 34], and have found wide applicability in various fields [39, 1, 19, 81, 83, 84, 12, 89, 37]. A GAN often consists of a generator $G$ and a discriminator $D$, with both trained adversarially with objective

$$\min_G \max_D \mathbb{E}_{\boldsymbol{x} \sim q_{\text{data}}(\boldsymbol{x})} \big[ \log D(\boldsymbol{x}) \big] + \mathbb{E}_{\boldsymbol{z} \sim p(\boldsymbol{z})} \big[ \log(1 - D(G(\boldsymbol{z}))) \big], \tag{1}$$

where $p(\boldsymbol{z})$ is a simple distribution (*e.g.*, Gaussian) and $q_{\text{data}}(\boldsymbol{x})$ is the underlying (unknown) data distribution from which we observe samples.

**Style-transfer techniques** An extensive literature [23, 71, 79, 13, 62, 38] has explored how one can manipulate the style of an image (*e.g.,* the texture [18, 30, 41] or attributes [33, 34]) by modulating the statistics of its latent features. These methods use style-transfer techniques like conditional instance normalization [18] or adaptive instance normalization [30], most of which are related to Feature-wise Linear Modulation (FiLM) [64]. FiLM imposes simple element-wise affine transformations to latent features of a neural network, showing remarkable effectiveness in various domains [30, 17, 33, 58]. Given a $d$-dimensional feature $\boldsymbol{h} \in \mathbb{R}^d$ from a layer of a neural network,[3] FiLM yields

$$\hat{\boldsymbol{h}} = \boldsymbol{\gamma} \odot \boldsymbol{h} + \boldsymbol{\beta}, \tag{2}$$

where $\hat{\boldsymbol{h}}$ is forwarded to the next layer, $\odot$ denotes the Hadamard product, and the scale $\boldsymbol{\gamma} \in \mathbb{R}^d$ and shift $\boldsymbol{\beta} \in \mathbb{R}^d$ may be conditioned on other information [18, 64, 17]. Different from FiLM modulating latent features, another technique named adaptive filter modulation (AdaFM) modulates source convolutional (Conv) filters to manipulate its "style" to deliver a boosted transfer performance [98]. Specifically, given a Conv filter $\mathbf{W} \in \mathbb{R}^{C_{\text{out}} \times C_{\text{in}} \times K_1 \times K_2}$, where $C_{\text{in}}/C_{\text{out}}$ denotes the number of input/output channels and $K_1 \times K_2$ is the kernel size, AdaFM yields

$$\hat{\mathbf{W}} = \boldsymbol{\Gamma} \odot \mathbf{W} + \mathbf{B}, \tag{3}$$

where the scale matrix $\boldsymbol{\Gamma} \in \mathbb{R}^{C_{\text{out}} \times C_{\text{in}}}$, the shift matrix $\mathbf{B} \in \mathbb{R}^{C_{\text{out}} \times C_{\text{in}}}$, and the modulated $\hat{\mathbf{W}}$ is used to convolve with input feature maps for output ones.

## 4 Proposed method

Targeting the fundamental memory issue of lifelong learning, we propose to exploit popular GANs to design a realistic *generative* memory (named GAN memory) to sequentially remember data-generating processes. Specifically, we consider a lifelong generation problem: the GAN memory

sequentially accesses a stream of datasets/tasks $\{\mathcal{D}_1, \mathcal{D}_2, \cdots\}$[4] (during task $t$, only $\mathcal{D}_t$ is accessible); after task $t$, the GAN memory should be able to synthesize realistic samples resembling $\{\mathcal{D}_1, \cdots, \mathcal{D}_t\}$. Below we first reveal a surprising discovery that lays the foundation of the paper. We then build on top of it our GAN memory followed by a detailed analysis, and finally compression techniques are presented to facilitate our GAN memory for lifelong problems with a long task sequence.

## 4.1 A surprising discovery

Moving well beyond the style-transfer literature modulating image features to manipulate its style [18, 30, 17], we discover that one can even modulate the "style" of a source generative/discriminative process (*e.g.,* a GAN generator/discriminator trained on a source dataset $\mathcal{D}_0$) to form synthesis power for a perceptually-distant target domain (*e.g.,* a generative/discriminative power on $\mathcal{D}_1$), via manipulating its FC and Conv layers with the style-modulation techniques developed below. Note different from the classical style-transfer literature, the "style" terminology here is associated with the characteristics of a function (*e.g.,* for a GAN generator, its style manifests as the generation content); because of the similarity in mathematics, we reuse that terminology but name our approach as style-modulation techniques.

Before introducing the technical details, we emphasize our basic assumption of well-behaved source FC and Conv parameters; often parameters from a GAN model trained on large-scale datasets satisfy that assumption, as discussed in the Introduction. To highlight our discovery, we choose a moderately sophisticated GP-GAN model [49] trained on the CelebA [43] (containing only faces) as the source,[5] and select perceptually-distant target datasets including Flowers, Cathedrals, Cats, Brain-tumor images, Chest X-rays, and Anime images (see Figure 5 and Section 5.1). With the style-modulation techniques detailed below, we observe realistic generations in all target domains (see Figure 5), even though the generation power is modulated from an entirely different source domain. Alternatively, given a specific target domain, that observation also implies the flexibility in choosing a source model (see also Appendix H), *i.e.,* the source (with well-behaved parameters) should have the same target data type, but it need not be related to the target domain. In the context of image-based data, this implies a certain universal structure to images, that may be captured within a GAN by one (relatively large) image dataset. Via appropriate style modulation of this model, it can be adapted to new and very different image classes, using potentially limited observations from those target domains.

We next present the style-modulation techniques employed here, modified FiLM (mFiLM) and modified AdaFM (mAdaFM), for modulating FC and Conv layers, respectively.

**FC layers** Given a source FC layer $h^{\text{source}} = \mathbf{W}z + b$ with weight $\mathbf{W} \in \mathbb{R}^{d_{\text{out}} \times d_{\text{in}}}$, bias $b \in \mathbb{R}^{d_{\text{out}}}$, and input $z \in \mathbb{R}^{d_{\text{in}}}$, mFiLM modulates its *parameters* to form a target function $h^{\text{target}} = \hat{\mathbf{W}}z + \hat{b}$ with

$$\hat{\mathbf{W}} = \boldsymbol{\gamma} \odot \frac{\mathbf{W} - \boldsymbol{\mu}}{\boldsymbol{\sigma}} + \boldsymbol{\beta}, \qquad \hat{b} = b + b_{\text{FC}}, \tag{4}$$

where $\boldsymbol{\mu}, \boldsymbol{\sigma} \in \mathbb{R}^{d_{\text{out}}}$, with the elements $\boldsymbol{\mu}_i, \boldsymbol{\sigma}_i$ denoting the mean and standard derivation of the vector $\mathbf{W}_{i,:}$, respectively; $\boldsymbol{\gamma}, \boldsymbol{\beta}, b_{\text{FC}} \in \mathbb{R}^{d_{\text{out}}}$ are target-specific scale, shift, and bias style parameters trained with target data ($\mathbf{W}$ and $b$ are frozen – from learning on the original source domain – during target training). One may interpret mFiLM as applying FiLM [64] (or batch normalization [31]) to a source FC *weight* to modulate its (row) statistics/style (encoded in $\boldsymbol{\mu}$ and $\boldsymbol{\sigma}$) to adapt to a target domain.

**Conv layers** Given a source Conv layer $\mathbf{H}^{\text{source}} = \mathbf{W} * \mathbf{H}' + b$ with input feature maps $\mathbf{H}'$, Conv filters $\mathbf{W} \in \mathbb{R}^{C_{\text{out}} \times C_{\text{in}} \times K_1 \times K_2}$, and bias $b \in \mathbb{R}^{C_{\text{out}}}$, we leverage mAdaFM to modulate its parameters to form a target Conv layer as $\mathbf{H}^{\text{target}} = \hat{\mathbf{W}} * \mathbf{H}' + \hat{b}$, where

$$\hat{\mathbf{W}} = \boldsymbol{\Gamma} \odot \frac{\mathbf{W} - \mathbf{M}}{\mathbf{S}} + \mathbf{B}, \qquad \hat{b} = b + b_{\text{Conv}}, \tag{5}$$

where $\mathbf{M}, \mathbf{S} \in \mathbb{R}^{C_{\text{out}} \times C_{\text{in}}}$ with the elements $\mathbf{M}_{i,j}, \mathbf{S}_{i,j}$ denoting the mean and standard derivation of the vector $\text{vec}(\mathbf{W}_{i,j,:,:})$, respectively. The trainable target-specific style parameters are $\boldsymbol{\Gamma}, \mathbf{B} \in$

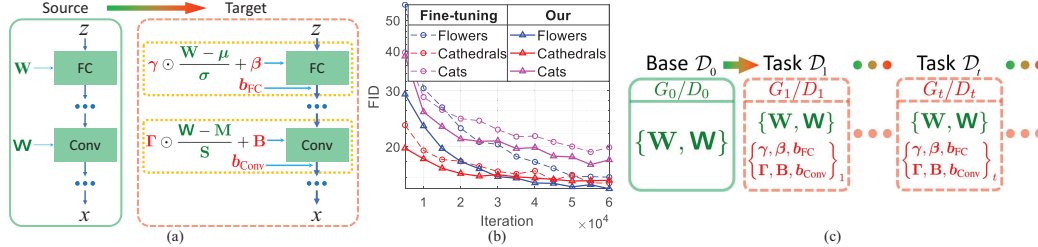

Figure 1: (a) Style modulation of a source GAN model (demonstrated with the generator, but it is also applied to the discriminator). Source parameters (green $\{\mathbf{W}, \mathbf{W}\}$) are frozen, with limited trainable style parameters (*i.e.,* red $\{\boldsymbol{\gamma}, \boldsymbol{\beta}, \boldsymbol{b}_{\text{FC}}, \boldsymbol{\Gamma}, \mathbf{B}, \boldsymbol{b}_{\text{Conv}}\}$) introduced to form the augmentation to the target domain. (b) Comparing our style modulation to the strong fine-tuning baseline (see Appendix B for details). (c) The architecture of our GAN memory for a stream of target generation tasks.

$\mathbb{R}^{C_{\text{out}} \times C_{\text{in}}}$ and $\boldsymbol{b}_{\text{Conv}} \in \mathbb{R}^{C_{\text{out}}}$, with $\mathbf{W}$ and $\boldsymbol{b}$ frozen. Similar to mFiLM, mAdaFM first removes the source style (encoded in $\mathbf{M}$ and $\mathbf{S}$), followed by leveraging $\boldsymbol{\Gamma}$ and $\mathbf{B}$ to learn the target style.

The adopted style modulation process (shown with the generator) is illustrated in Figure 1(a). Given a source GAN model, we transfer and freeze all its parameters to a target domain, followed by using mFiLM/mAdaFM in (4)/(5) to modulate its FC/Conv layers (with style parameters $\{\boldsymbol{\gamma}, \boldsymbol{\beta}, \boldsymbol{b}_{\text{FC}}, \boldsymbol{\Gamma}, \mathbf{B}, \boldsymbol{b}_{\text{Conv}}\}$) to yield the target generation model. With that style modulation, we transfer the source knowledge (within the frozen parameters) to the (potentially) perceptually-distant target domain to facilitate a realistic generation therein (see Figure 5 for generated samples). For quantitative evaluations, comparisons are made with a strong baseline, *i.e.,* fine-tuning the whole source model (including both generator and discriminator) on target data, which is expected to outperform training from scratch in both efficiency and performance by referring to [85] and the transfer learning literature. The FID scores (lower is better) [29] from both methods are summarized in Figure 1(b), where our method consistently outperforms that strong baseline by a large margin, on both training efficiency and performance,[6] highlighting the valuable knowledge within frozen source parameters (apparently a type of universal information about images) and showing a potentially better way for transfer learning.

Complementing the common knowledge of transfer learning, *i.e.,* low-level filters (those close to observations) are generally applicable, while high-level ones are task-specific [90, 93, 44, 4, 5, 20, 58, 98], the above discovery reveals an orthogonal dimensionality for transfer learning. Specifically, the shape of kernels (*i.e.,* relative relationship among kernel elements $\{\mathbf{W}_{i,j,1,1}, \mathbf{W}_{i,j,1,2}, \cdots\}$) may be generally transferable whereas the *statistics* of kernels (the mean $\mathbf{M}_{i,j}$ or standard derivation $\mathbf{S}_{i,j}$) or among-kernel correlations (*e.g.,* relative relationship among kernel statistics) are task-specific. A similar conjecture on low-level Conv filters was discussed in [98]; we reveal such patterns even hold for the whole GAN model (for both low-level and high-level kernels of the generator/discriminator), which is unanticipated because common experience associates high-level kernels with task-specific information. This insight might reveal a new avenue for transfer learning.

### 4.2   GAN memory to sequentially remember a stream of generative processes

Based on the above observations, we propose a GAN memory that has the power to realistically remember a stream of generative processes with *no* forgetting. The key observation here is that when modulating a source GAN model for a target domain, the source model is frozen (thus no forgetting of the source) with a limited set of target-specific style parameters (*i.e.,* no influence among tasks) introduced to form a realistic target generation. Accordingly, we can use a well-behaved source GAN model as a base, followed by sequentially modulating its "style" to deliver realistic generation power on a stream of target datasets,[7] as illustrated in Figure 1(c). We use the same settings as in Section

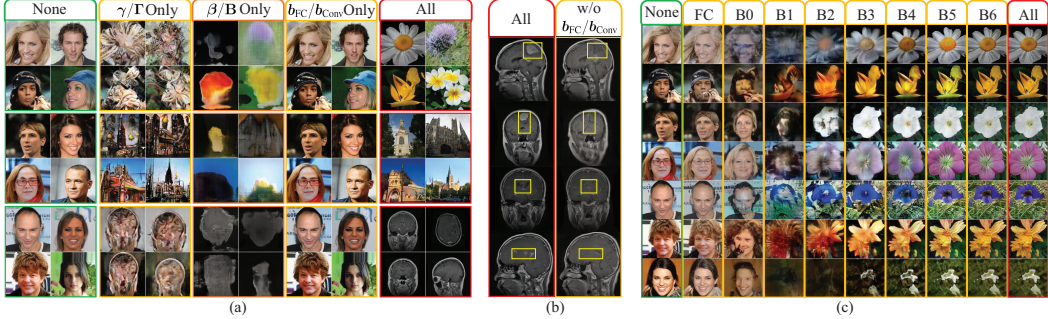

Figure 2: (a) Style parameters modulate different generation perspectives. (b) The biases model sparse objects. (c) Modulations in different blocks have different strength/focus over the generation. B$m$ is the $m$th residual block. B0/B6 is closest to the noise/observation. See Appendix C for details.

4.1. As style parameters are often limited (and can be further compressed as in Section 4.3), one could expect from our GAN memory a substantial compression of a stream of datasets, while not forgetting realistic generative replay (see Figure 5). To help better understand how/why our GAN memory works, we next reveal five of its properties.

*Each group of style parameters modulates a different generation perspective.* Style parameters consist of three groups, *i.e.,* scales $\{\gamma, \Gamma\}$, shifts $\{\beta, \mathbf{B}\}$, and biases $\{b_{\text{FC}}, b_{\text{Conv}}\}$. Taking as examples the style parameters trained on the Flowers, Cathedrals, and Brain-tumor images, Figure 2(a) demonstrates the generation perspective modulated by each group: ($i$) when none/all groups are applied, GAN memory generates realistic source/target images (see the first/last column, respectively); ($ii$) when only modulating via the scales (denoted as $\gamma/\Gamma$ Only, the second column), the generated samples show textural/structural information from target domains (like the textures on petals or the contours of buildings/skulls); ($iii$) as shown in the third column, the shifts if solely applied principally control the low-frequency color information from target domains; ($iv$) finally the biases (see the forth column) control the illumination and localized objects (not obvious here). To clearly reveal the role played by the biases, we keep both scales and shifts fixed, and compare the generated samples with/without biases on the Brain-tumor dataset; Figure 2(b) shows the biases are important in modeling localized objects like the tumor or tissue details, which may be valuable in pathological analysis. Also the following Figure 3(a) shows that the biases help with a better training efficiency.

*Style parameters within different blocks show different strength/focus over the generation.* Figure 2(c) shows the generated samples on the Flowers dataset when gradually and accumulatively adding modulations to each block (from FC to B6). To begin with, the FC modulation changes the overall contrast and illumination of the generation; then the style parameters in B0-B3 make the most effort to modulate the face manifold into a flower, followed by modulations in B4-B6 refining the generation details. Such patterns are somewhat consistent with existing practice, in the sense that high-level/low-level *kernel statistics* are more task-specific/generally-applicable. It's therefore interesting to consider combining the two orthogonal dimensions, *i.e.,* the existing low-layer/high-layer split and the revealed kernel-shape/kernel-statistics split, for potentially better transfer learning.

*Normalization contributes significantly to better training efficiency and performance.* To investigate how the weight normalization and the biases in (4)/(5) contribute, ablation studies are conducted, with the results shown in Figure 3(a). With the weight normalization to remove the source "style," our method shows both an improved training efficiency and a boosted performance; the biases contribute to a better efficiency but with minimal influence on the performance.

*GAN memory enables smooth interpolations among generative processes*, by dynamically combining two (or more) sets of style parameters. Figure 3(b) demonstrates smooth interpolations between flower and cat generative processes. Such a property can be used to deliver versatile data augmentation among different domains, which may, for example, benefit a downstream robust classification [61, 7].

*GAN memory readily generalizes to label-conditioned generation problems*, where each task dataset $\mathcal{D}_t$ contains both observations $\{\boldsymbol{x}\}$ and the paired labels $\{y\}$ with $y \in \{1, \cdots, C_t\}$. Mimicking the conditional GAN [50], we specify one FC bias per class to model the label information; accordingly, the style parameters for the $t$th task are $\{\boldsymbol{\gamma}, \boldsymbol{\beta}, \{\boldsymbol{b}_{\text{FC}}\}_{i=1}^{C_t}, \boldsymbol{\Gamma}, \mathbf{B}, \boldsymbol{b}_{\text{Conv}}\}$. Figure 3(c) shows the realistic

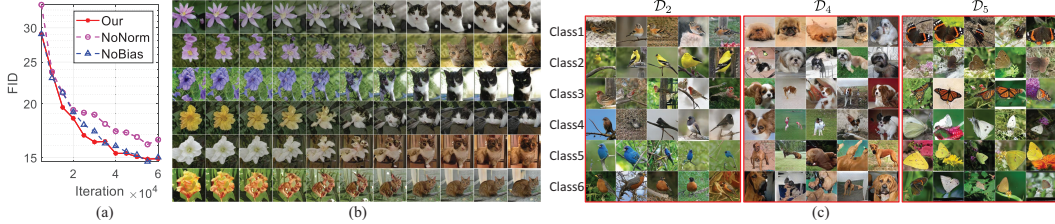

Figure 3: (a) Ablation study on Flowers. (b) Smooth interpolations between flower and cat generative processes. (c) Realistic replay from our conditional-GAN memory. See Appendix D for details and more demonstrations.

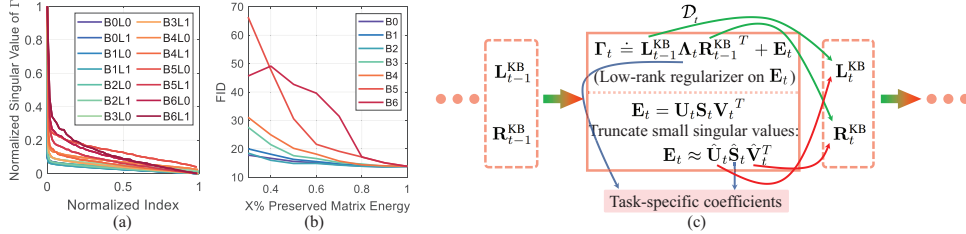

Figure 4: (a) Normalized singular values of $\mathbf{\Gamma}$ at all blocks/layers on Flowers (see Appendix Figure 15 for $\mathbf{B}$). Maximum normalization is applied to both axes. B$m$L$n$ stands for the $n$th Conv layer of the $m$th block. (b) Influence of truncation (via preserving $X\%$ matrix energy [56]) on generation. (c) GAN memory with further compression and knowledge sharing. See Appendix G for details.

replay from our conditional-GAN memory after sequential training on five tasks (exampled with bird, dog, and butterfly; see Section 5.2 for details).

## 4.3 GAN memory with further compression

Delivering realistic sequentially learned generations with no forgetting, the GAN memory presented above is expected to be sufficient for many practical applications with a moderate number of tasks. However, for challenging situations with many tasks, to save a set of style parameters for each task might become prohibitive.[8] For that problem, we reveal below ($i$) style parameters (*i.e.,* the expensive matrices $\mathbf{\Gamma}$ and $\mathbf{B}$) can be compressed to lower the memory cost for each task; ($ii$) one can exploit sharing parameters among tasks to further enhance memory savings.[9]

We first investigate the singular values of $\mathbf{\Gamma}$ and $\mathbf{B}$ learned at different blocks/layers, and summarize them in Figure 4(a). It's clear the $\mathbf{\Gamma}$ and $\mathbf{B}$ parameters are in general low-rank; moreover, the closer a $\mathbf{\Gamma}$ or $\mathbf{B}$ is to the noise (often with a larger matrix size and thus more expensive), the stronger its low-rank property (yielding better compressibility). Accordingly, we truncate/zero-out small singular values at each block/layer to test the corresponding performance decline. Figure 4(b) summarizes the results, where keeping $80\%$ matrix energy [56] ($\approx 35\%$ top singular values) of $\mathbf{\Gamma}$ and $\mathbf{B}$ in B0-B4 almost has the same performance, verifying the compressibility of $\mathbf{\Gamma}$ and $\mathbf{B}$.

Based on the compressibility of $\mathbf{\Gamma}$ and $\mathbf{B}$, we next reveal parameter sharing among tasks can be exploited for further memory saving. Specifically, we propose to leverage matrix factorization and low-rank regularization to form a lifelong knowledge base mimicking [68]. Taking the $t$th task as an example, instead of optimizing over a task-specific $\mathbf{\Gamma}_t$ (similarly for $\mathbf{B}_t$), we alternatively optimize over its parameterization $\mathbf{\Gamma}_t \doteq \mathbf{L}_{t-1}^{\text{KB}}\mathbf{\Lambda}_t(\mathbf{R}_{t-1}^{\text{KB}})^T + \mathbf{E}_t$, where $\mathbf{L}_{t-1}^{\text{KB}}$ and $\mathbf{R}_{t-1}^{\text{KB}}$ are respectively the existing left/right knowledge base, $\mathbf{\Lambda}_t = \text{Diag}(\boldsymbol{\lambda}_t)$, and $\boldsymbol{\lambda}_t$ and $\mathbf{E}_t$ are task-specific trainable parameters. The nuclear norm $\|\mathbf{E}_t\|_*$ is added to the loss to encourage a low-rank property. After training on the $t$th task, we apply singular value decomposition to $\mathbf{E}_t$, keep the top singular values to preserved $X\%$ matrix energy, and use the corresponding left/right singular vectors to update the left/right knowledge base to $\mathbf{L}_t^{\text{KB}}$ and $\mathbf{R}_t^{\text{KB}}$. The overall procedure is demonstrated in Figure 4(c).

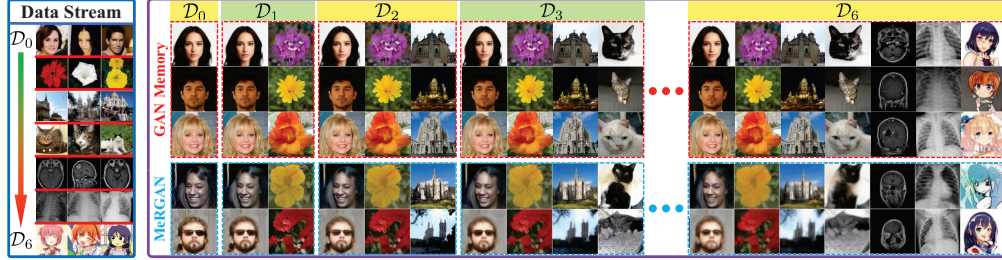

Figure 5: The task/dataset stream (left) and generated samples after training on each task/dataset (right).

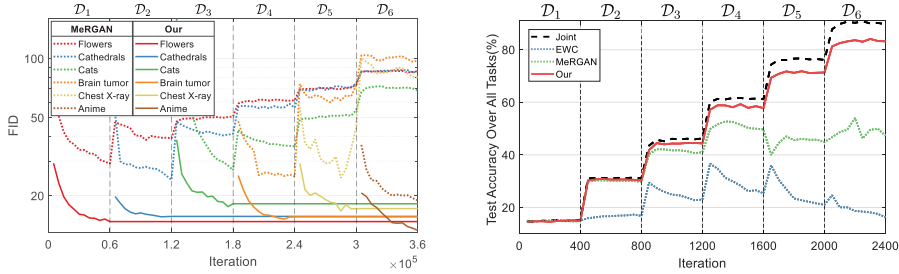

Figure 6: (Left) FID curves on the lifelong generation problem of Section 5.1. (Right) Classification accuracy on the lifelong classification problem of Section 5.2. The curve labeled "Joint" denotes the upper-bound, where the classifier is trained jointly on all the data from the current and historical tasks.

## 5 Experiments

Experiments on high-dimensional image datasets from diverse fields are conducted to demonstrate the effectiveness of the proposed techniques. Specifically, we first test our GAN memory to realistically remember a stream of generative processes; we then show that our (conditional) GAN memory can be used to form realistic pseudo rehearsal (synthesis of data from prior tasks) to alleviate catastrophic forgetting for challenging lifelong classification tasks; and finally for long task sequences, we reveal the techniques from Section 4.3 enable significant memory savings but with comparable performance. Detailed experimental settings are given in Appendix A.

### 5.1 GAN memory on a stream of generation tasks

To demonstrate the superiority of our GAN memory over existing replay-based methods, we design a challenging lifelong generation problem consisting of 6 perceptually-distant tasks/datasets (see Figure 5): Flowers [57], Cathedrals [99], Cats [97], Brain-tumor images [15], Chest X-rays [35], and Anime faces.[10] The GP-GAN [49] trained on the CelebA [43] ($\mathcal{D}_0$) is selected as the base; other well-behaved GAN models may readily be considered. We compare our GAN memory with the memory replay GAN (MeRGAN) [86], which keeps another copy of the generator in memory to replay historical generations, to mitigate catastrophic forgetting. Qualitative comparisons between both methods along the sequential training are shown in Figure 5. It's clear our GAN memory delivers realistic generations with no forgetting on historical tasks, whereas MeRGAN shows increasingly blurry historical generations with reinforced generation artifacts [85] as training proceeds. For quantitative comparisons, the FID scores [29] along the training are summarized in Figure 6 (left), highlighting the advantages of our GAN memory, *i.e.,* realistic generations with no forgetting. Also revealed is that our method even performs better with a better efficiency for the current task, likely thanks to the transferred common knowledge within frozen base parameters.

### 5.2 Conditional-GAN memory as pseudo rehearsal for lifelong classifications

Witnessing the success of our (conditional) GAN memory in continually remembering realistic generations, we next utilize it as a pseudo rehearsal to assist downstream lifelong classifications. Specifically, we design 6 streaming tasks by selecting fish, bird, snake, dog, butterfly, and insect

images from the ImageNet [67]; each task is then formalized as a 6-classification problem (*e.g.,* for task bird, the goal is to classify 6 categories of birds). We employ the challenging class-incremental learning setup [78], *i.e.,* the classifier (after task $t$) is expected to accurately classify all observed (first $6t$) categories. For comparisons, we employ the regularization-based EWC [36] and the generative-replay-based MeRGAN [86]. For replay-based methods (*i.e.,* the MeRGAN and our conditional-GAN memory), at task $t$, we train the classifier with a combined dataset that contains both the current observed data (from task $t$) and the generated/replayed historical samples (mimicking the data from task $1 \sim t - 1$); after that, the MeRGAN/conditional-GAN-memory is updated to remember the current data generative process [72, 86]. See Appendix F for more details.

Testing classification accuracy on all 36 categories from the compared methods along training are summarized in Figure 6 (right), It's clear that EWC barely works in the class-incremental learning scenario [77, 40, 78]. MeRGAN doesn't work well when the task sequence is long, because of its increasingly blurry rehearsal as shown in Figure 5. By comparison, our conditional-GAN memory with no forgetting succeeds in stably maintaining an increasing accuracy as new tasks come and shows performance close to the joint-training upper-bound, highlighting its practical value in alleviating catastrophic forgetting for general lifelong learning problems. See Appendix F for evolution of the performance on each task along the training process.

## 5.3 GAN memory with parameter compression and sharing

Table 1: Comparisons of GAN memory with (Compr) or without (Naive) compression techniques. #Params denotes the number of newly-introduced style parameters for each task. The number of the frozen source parameters is 52.2M.

| Task | $\mathcal{D}_1$ | $\mathcal{D}_2$ | $\mathcal{D}_3$ | $\mathcal{D}_4$ | $\mathcal{D}_5$ | $\mathcal{D}_6$ |
|---|---|---|---|---|---|---|
| #Params$_{\text{Naive}}$ | 10.6M | 10.6M | 10.6M | 10.6M | 10.6M | 10.6M |
| #Params$_{\text{Compr}}$ | 3.8M | 1.7M | 0.9M | 0.8M | 0.3M | 0.3M |
| #Params$_{\text{Compr}}$/#Params$_{\text{Naive}}$ | 36.4% | 16.0% | 9.0% | 7.6% | 2.7% | 2.9% |
| FID (Compr) | **27.67** | 23.49 | **28.90** | **31.07** | 32.19 | 49.28 |
| FID (Naive) | 28.89 | **22.80** | 34.36 | 35.72 | **29.50** | **40.03** |

To verify the effectiveness of the compression techniques presented in Section 4.3, which are believed valuable for lifelong learning with many tasks, we design another lifelong generation problem based on the ImageNet for better demonstration.[11] Specifically, we select 6 categories of butterfly images to form a stream of 6 generation tasks/datasets (one category per task), among which similarity/knowledge-sharability is expected. The procedure shown in Figure 4(c) is employed for our method. See Appendix G for details. We compare our GAN memory with compression techniques (denoted as Compr) to its naive implementation with task-specific style parameters (Naive), with the results summarized in Table 1. It's clear that ($i$) even for task $\mathcal{D}_1$ (with an empty knowledge base), the low-rank property of $\mathbf{\Gamma}/\mathbf{B}$ enables a significant parameter compression; ($ii$) based on the existing knowledge base, much less new parameters are necessary to form a new generation model (*e.g.,* for task $\mathcal{D}_2$ or $\mathcal{D}_3$), confirming the reusability/sharability of existing knowledge; and ($iii$) though it has significant parameter compression, Compr delivers comparable performance to Naive, confirming the effectiveness of the presented compression techniques.

## 6 Conclusions

We reveal that one can modulate the "style" of a GAN model to accurately synthesize the statistics of data from perceptually-distant targets. Based on that recognition, we propose our GAN memory with growing generation power, but with no forgetting. We then analyze our GAN memory in detail, reveal for it new compression techniques, and empirically verify its advantages over existing methods. Concerning a better base model, one may leverage the generative replay ability of the GAN memory to form a long-period update, mimicking human behavior during rapid-eye-movement sleep [75, 22].

## Broader impact

Capable of remembering a stream of data generative processes with no forgetting, our GAN memory has the following potential positive impact in the society: ($i$) it may serve as a powerful generative replay for challenging lifelong applications such as self-driving; ($ii$) as no original data are saved, the concerns on data privacy may be well addressed; ($iii$) GAN memory enables flexible control over the replayed contents, which is of great value to practical applications, where training data are unbalanced, or where one needs to flexibly select which model capability to maintain/forget during training; ($iv$) the counter-intuitive discovery that lays the foundation of our GAN memory may disclose another dimension for transfer learning, *i.e.,* the kernel shape is generally applicable while the corresponding kernel statistics/style is task-specific; similar patterns may also apply to classifiers. Since our GAN memory is built on top of GANs, it may inherit their ethical and societal impact. Despite being versatility, GANs may be improperly used to synthesize fake images/news/videos, resulting in negative consequences. Furthermore, we should be cautious of the failure of adversarial training due to mode collapse, which may compromise the generative capability on the current task. Note that training failure, if it happens, will not hurt the performance on other tasks, showing certain robustness of our GAN memory.

## Acknowledgements

We thank the anonymous reviewers for their constructive comments. The research was supported in part by DARPA, DOE, NIH, NSF, ONR and SOC R&D lab of Samsung Semiconductor Inc. The Titan Xp GPU used was donated by the NVIDIA Corporation.

## Footnotes

[2] One can of course expect better performance if a better source model (pretrained on a large-scale dense and diverse dataset) is used.

[3]For simplicity, we omit layer-index notation throughout the paper.

[4]This setup is not limited, as it's often convenient to use a physical memory buffer to form the dataset stream. Note concerning practical applications, it's often unnecessary to consider the extreme case where each dataset $\mathcal{D}_t$ contains only one data sample; accordingly, we assume a moderate number of samples per dataset by default.

[5]See Appendix A for the detailed architectures. Note our method is deemed considerably robust to the (pretrained) source model, as discussed in Appendix H, where additional experiments are conduced based on a different source GAN model pretrained on LSUN Bedrooms [91].

[6] The newly-introduced style parameters $\{\boldsymbol{\gamma}, \boldsymbol{\beta}, \boldsymbol{b}_{\text{FC}}, \boldsymbol{\Gamma}, \mathbf{B}, \boldsymbol{b}_{\text{Conv}}\}$ of our method are only about 20% of those of the fine-tuning baseline, yet they deliver better efficiency and performance. This is likely because the target data are not sufficient enough to train well-behaved parameters like the source ones, when performing fine-tuning alone. Note that with the techniques from Section 4.3, one can use much less style parameters (*e.g.,* 7.3% of those of the fine-tuning baseline) to yield a comparable performance.

[7] Our GAN memory is amenable to streaming training, parallel training, and even their flexible combinations, thanks to the frozen base model and task-specific style parameters.

[8] A compromise may save limited task-specific style parameters to hard disks and only load them when used.

[9] The cheap vector parameters $\{\boldsymbol{\gamma}, \boldsymbol{\beta}, \boldsymbol{b}_{\text{FC}}, \boldsymbol{b}_{\text{Conv}}\}$ can be similarly processed in a dictionary learning manner. As they are often quite inexpensive to retain, we consider them being task-specific for simplicity.

[10]https://github.com/jayleicn/animeGAN

[11] The datasets from Section 5.1 are too perceptually-distant to illustrate parameter sharability among tasks.

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
