[Supplementary Material]

# Appendix of GAN Memory with No Forgetting

**Yulai Cong, Miaoyun Zhao, Jianqiao Li, Sijia Wang, Lawrence Carin**
**Department of ECE, Duke University**

## A   Experimental settings

For all experiments about GAN memory and MeRGAN, we inherit the architecture and experimental settings from GP-GAN [49]. Note, for both GAN memory and MeRGAN, we use the GP-GAN model pretrained on CelebA. The architecture for the generator is shown in Figure 7, where we denote the $m$th residual block as B$m$ and the $n$th convolutional layer within residual block as L$n$. In the implementation of GAN memory, we apply style modulation on all layers except the last Conv layer for generator, and apply the proposed style modulation on all layers except the last FC layer for discriminator. Adam is used as the optimizer with learning rate $1 \times 10^{-4}$ and coefficients $(\beta_1, \beta_2) = (0.0, 0.99)$. For the discriminator, gradient penalty on real samples ($R_1$-regularizer) is applied with $\gamma = 10.0$. For MeRGAN, the Replay alignment parameter is set as $\lambda_{\text{RA}} = 1 \times 10^{-3}$, which is the same as their publication.

Figure 7: (a) The generator architecture adopted in this paper. (b) The detailed architecture of the residual block.

All images are resized to $256 \times 256$ for consistency. The dimension of the input noise is set to 256 for all tasks. The FID scores are calculated using N real and generated images, for the dataset with less than 10,000 images, we set N as the number of data samples; for the dataset with larger than 10,000 samples, we set N as 10,000.

## B   On Figure 1(b)

To demonstrate that the introduced trainable parameters in our GAN memory is enough to manage a decent performance on new task. We test its performance via FID and compare with a strong baseline Fine-tuning.

On the Fine-tuning method. It inherits the architecture from GP-GAN which is the same as the green/frozen part of our GAN memory (see Figure 1(a)). Given a target task/data (*e.g.,* Flowers, Cathedrals, or Cats), all the parameters are trainable and fine-tuned to fit the target data. We consider Fine-tuning as a strong baseline because it utilizes the whole model power on the target data.

## C   On Figure 2

For all illustrations, we train GAN memory on the target data and record the well trained style parameters as $\{\gamma, \beta, b_{\text{FC}}, \Gamma, \mathbf{B}, b_{\text{Conv}}\}_{t=1}$. According to equation (4) and (5), we can represent the style parameters for the source data CelebA as $\{\gamma, \beta, b_{\text{FC}}, \Gamma, \mathbf{B}, b_{\text{Conv}}\}_{t=0} = \{\mu, \sigma, \mathbf{0}, \mathbf{M}, \mathbf{S}, \mathbf{0}\}$. Then, we can get the generation by selectively replacing $\{\gamma, \beta, b_{\text{FC}}, \Gamma, \mathbf{B}, b_{\text{Conv}}\}_{t=0}$ with $\{\gamma, \beta, b_{\text{FC}}, \Gamma, \mathbf{B}, b_{\text{Conv}}\}_{t=1}$. Note, for Figure 2(a)(b) the operations are explained within one specific FC/Conv layer, one need to apply it to all layers in real implementation. The detailed techniques are as follows.

*On Figure 2(a).* Take the target data Flowers as an example,

- "None" means no modulation from target data is applied, we only use the modulation from the source data $\{\mu, \sigma, \mathbf{0}, \mathbf{M}, \mathbf{S}, \mathbf{0}\}$ and get face images;
- "$\gamma/\Gamma$ Only" means that we replace the $\gamma/\Gamma$ parameters from source data with the one from target data, namely using the modulation $\{\{\gamma\}_{t=1}, \sigma, \mathbf{0}, \{\Gamma\}_{t=1}, \mathbf{S}, \mathbf{0}\}$ for generation;
- "$\beta/\mathbf{B}$ Only" means that we replace the $\beta/\mathbf{B}$ parameters from source data with the one from target data, namely using the modulation $\{\mu, \{\beta\}_{t=1}, \mathbf{0}, \mathbf{M}, \{\mathbf{B}\}_{t=1}, \mathbf{0}\}$ for generation;
- "$b_{\text{FC}}/b_{\text{Conv}}$ Only" means that we replace the $b_{\text{FC}}/b_{\text{Conv}}$ parameters from source data with the one from target data, namely using the modulation $\{\mu, \sigma, \{b_{\text{FC}}\}_{t=1}, \mathbf{M}, \mathbf{S}, \{b_{\text{Conv}}\}_{t=1}\}$ for generation;
- "All" means all style parameters from target data $\{\gamma, \beta, b_{\text{FC}}, \Gamma, \mathbf{B}, b_{\text{Conv}}\}_{t=1}$ are applied and the model generates flowers;

*On Figure 2(b).*

- "All" is obtained via a similar way to that of Figure 2(a);
- "w/o $b_{\text{FC}}/b_{\text{Conv}}$" means using the style parameters from target data without $b_{\text{FC}}/b_{\text{Conv}}$, namely using the modulation $\{\{\gamma\}_{t=1}, \{\beta\}_{t=1}, \mathbf{0}, \{\Gamma\}_{t=1}, \{\mathbf{B}\}_{t=1}, \mathbf{0}\}$ for generation;

*On Figure 2(c).*

- "None" and "All" are obtained via a similar way to that of Figure 2(a);
- "FC" is obtained by applying a newly designed style parameters which copies the style parameters for source data and replaces these style parameters within FC layer with the style parameters within the FC layer for target data;
- "B0" is obtained by copying the designed style parameters under the "FC" setting and replacing these style parameters within B0 block with those from target data;
- "B1" is obtained by copying the designed style parameters under the "B0" setting and replacing these style parameters within B1 block with those from target data;
- and so forth for the later blocks.

## D   On Figure 3

### D.1   On Figure 3(a)

The ablation study is conducted to test the effect the normalization operation and the bias term on GAN memory.

- "Our" is our GAN memory;
- "NoNorm" is a modified version of our GAN memory which removes the normalization on $\mathbf{W}/\mathsf{W}$ in Equation (4)/(5) and results in,

$$\hat{\mathbf{W}} = \gamma \odot \mathbf{W} + \beta$$

and

$$\hat{\mathsf{W}} = \Gamma \odot \mathsf{W} + \mathbf{B}$$

- "NoBias" is a modified version of our GAN memory which removes the bias term $b_{\text{FC}}/b_{\text{Conv}}$ in Equation (4)/(5) and results in,

$$\hat{b} = b$$

**D.2 On Figure 3(b)**

Here we discuss the detailed techniques for interpolation among different generative processes with our GAN memory and show more examples in Figure 8, 9, 10, and 11. Taking the smooth interpolation between flowers and cats generative processes as an example, we do the following procedures to get the results.

- Train GAN memory on Flowers and Cats independently and get the well trained style parameters for Flowers as $\{\boldsymbol{\gamma}, \boldsymbol{\beta}, \boldsymbol{b}_{\text{FC}}, \boldsymbol{\Gamma}, \mathbf{B}, \boldsymbol{b}_{\text{Conv}}\}_{t=1}$ and for Cats as $\{\boldsymbol{\gamma}, \boldsymbol{\beta}, \boldsymbol{b}_{\text{FC}}, \boldsymbol{\Gamma}, \mathbf{B}, \boldsymbol{b}_{\text{Conv}}\}_{t=2}$;
- Sample a bunch of random noise $\boldsymbol{z}$ (*e.g.,* 8 random noise vectors) and fix it;
- Get the interpolated generation by dynamically combining the two sets of style parameters via

$$(1 - \lambda_{\text{Interp}})\{\boldsymbol{\gamma}, \boldsymbol{\beta}, \boldsymbol{b}_{\text{FC}}, \boldsymbol{\Gamma}, \mathbf{B}, \boldsymbol{b}_{\text{Conv}}\}_{t=1} + \lambda_{\text{Interp}}\{\boldsymbol{\gamma}, \boldsymbol{\beta}, \boldsymbol{b}_{\text{FC}}, \boldsymbol{\Gamma}, \mathbf{B}, \boldsymbol{b}_{\text{Conv}}\}_{t=2},$$

with $\lambda_{\text{Interp}}$ varying from $0$ to $1$.

Note, the task CelebA has been well pretrained and we can obtain the style parameters directly as $\{\boldsymbol{\gamma}, \boldsymbol{\beta}, \boldsymbol{b}_{\text{FC}}, \boldsymbol{\Gamma}, \mathbf{B}, \boldsymbol{b}_{\text{Conv}}\}_{t=0} = \{\boldsymbol{\mu}, \boldsymbol{\sigma}, \mathbf{0}, \mathbf{M}, \mathbf{S}, \mathbf{0}\}$.

Figure 8: Smooth interpolations between faces and flowers generative processes via GAN memory.

Figure 9: Smooth interpolations between flowers and cats generative processes via GAN memory.

Figure 10: Smooth interpolations between cats and cathedrals generative processes via GAN memory.

Figure 11: Smooth interpolations between cathedrals and Brain-tumor images generative processes via GAN memory.

# E More results for Section 5.1

Given the same model pretrained for CelebA, we train GAN memory and MeRGAN on a sequence of tasks: Flowers (8,189 images), Cathedrals (7,350 images), Cats (9,993 images), Brain-tumor images (3,064 images), Chest X-rays (5,216 images), and Anime images (115,085 images). When task $t$ presents, only the data from $\mathcal{D}_t$ is available, and GAN memory/MeRGAN is expected to generate samples for all learned tasks $\mathcal{D}_1, \cdots, \mathcal{D}_t$. Due to the limited space, we only show part of the results in Figure 5. To make it clearer, Figure 12 shows a complete results with the generations for all tasks along the training process listed.

(a) GAN Memory

(b) MeRGAN

Figure 12: Comparing the generated samples from GAN memory (top) and MeRGAN (bottom) on lifelong learning of a sequential generation tasks: Flowers, Cathedrals, Cats, Brain-tumor images, Chest X-rays, and Anime images. MeRGAN shows an increasing blurriness along the task sequence, while GAN memory can learn to perform the current task and remembering the old tasks realistically.

## F  Experimental settings for lifelong classification

Given a sequence of 6 tasks: fish, bird, snake, dog, butterfly, and insect selected from ImageNet. Each task is a classification problem over six categories/sub-classes. We employ the challenging class-incremental learning setup, when task $t$ presents, the classifier (have been trained on previous $t-1$ tasks) is expected to accurately classify all observed $6t$ categories (namely, the previous $6(t-1)$ categories plus the current 6 categories). There are $6 \times 6 = 36$ categories/sub-classes in total, for each category/sub-class, we randomly select 1200 images for training and 100 for testing.

As for the classification model, we select the pretrained ResNet18 model for ImageNet. The optimizer for classification is selected as Adam with learning rate $1 \times 10^{-4}$ and coefficients $(\beta_1, \beta_2) = (0.9, 0.999)$.

For EWC, we adopt the code from [78], and set the parameters as $\lambda_{\text{EWC}} = 10^4$. We also tried the setting $\lambda_{\text{EWC}} = 10^9$ used in [86], however, the results shows that $\lambda_{\text{EWC}} = 10^9$ is too strong in our case, *e.g.,* when the learning proceeds to task $5/6$, the parameters are strongly pinned to the one learned for previous tasks, which makes it difficult to learn the new/current task, and the accuracy for the current task kept almost 0. For GAN memory and MeRGAN, we adopt the two step framework from [72], every time a new task $t$ presents, we $(i)$ train GAN memory/MeRGAN (which has already remembered all the previous tasks $1 \sim t-1$) to remember the generation for both the previous tasks and the current task; $(ii)$ at the same time, train the classifier/solver on a combined dataset: real samples for current task with their labels provided and generated samples for previous tasks replayed by GAN memory/MeRGAN. Since we apply label conditioned generation, where the category is an input, the replay process can be readily controlled with the reliable sampling of $(\boldsymbol{x}, \boldsymbol{y})$ pairs (*e.g.,* sample the label $\boldsymbol{y}$ and noise $\boldsymbol{z}$ first and then sample data $\boldsymbol{x}$ via the generator $\boldsymbol{x} = G(\boldsymbol{z}, \boldsymbol{y})$), which is considered effective in avoiding potential classification errors and biased sampling towards recent categories [86]. Note, for both GAN memory and MeRGAN, we used the GP-GAN model pretrained on CelebA.

The batch size for EWC is set as $n = 36$. For replay based method, the batch size for classification is set as follows. When task $t = 1, 2, \cdots, T$ presents, the batch size is $n \times t$: $(i)$ $n$ samples from the current task; $(ii)$ $n \times (t-1)$ generated samples replayed by GAN memory/MeRGAN for previous $t-1$ tasks (each task with $n$ replayed samples).

The classification accuracy shown in Figure 6 (right) is obtained by testing the classifier on the whole test dataset for all tasks with $36 \times 100 = 3600$ images. We also show in Figure 13 the evolution of the classification accuracy for each task during the whole training process. Take Figure 13(a) as an

example, the shown classification accuracy for task 1 on $\mathcal{D}_1$ is obtained by testing the classifier on the test images belonging to $\mathcal{D}_1$ with $6 \times 100 = 600$ images.

We observe from Figure 13 that, ($i$) EWC forgets the previous task much quickly than the others, *e.g.,* when the learning proceeds to task $\mathcal{D}_6$, the knowledge/capability learned from task $\mathcal{D}_1, \mathcal{D}_2$, and $\mathcal{D}_3$ are totally forgotten and the performances are seriously decreased; ($ii$) MeRGAN shows clear performance decline on historical classifications due to its blurry rehearsal on complex datasets, which is especially obvious when the task sequence becomes long (see Figure 13(a)(b)(c)). ($iii$) By comparison, our (conditional) GAN memory succeeds in stably maintaining historical accuracy even when the sequence becomes long thanks to its realistic pseudo rehearsal, highlighting its practical values in serving as a generative memory for general lifelong learning problems.

Figure 13: The evolution of the classification accuracy for each task during the whole training process: (a) Task $\mathcal{D}_1$; (b) Task $\mathcal{D}_2$; (c) Task $\mathcal{D}_3$; (d) Task $\mathcal{D}_4$; (e) Task $\mathcal{D}_5$; (f) Task $\mathcal{D}_6$.

Figure 14 shows the realistic replay from our conditional-GAN memory after sequential training on five tasks.[12]

(a) fish

(b) bird

(c) snake

(d) dog

(e) butterfly

Figure 14: Realistic replay from our conditional-GAN memory. (a) fish; (b) bird; (c) snake; (d) dog; (e) butterfly.

# G GAN memory with further compression

## G.1 The compression method

Our proposed GAN memory is practical when the sequence of tasks is mediate. However when the number of tasks is extremely large, *e.g.,* 1000, the introduced parameters for GAN memory will be expensive too. To make the GAN memory scalable to the number of tasks, we propose a naive method to compress the scale and bias $\Gamma, \mathbf{B}$ by taking advantage of the redundancy (Low-rank) within it, which is potential to remember extremely long sequence of tasks with finite parameters.

Similar to Figure 4(a), the singular values of scale $\Gamma$ and bias $\mathbf{B}$ learned at different blocks/layers are summarized in Figure 15. Obviously, those bias $\mathbf{B}$ are also generally low-rank and the closer a bias $\mathbf{B}$ to the noise, the stronger low-rank property it shows.

Maximum normalization is applied for both Figure 4(a) and Figure 15 to provide a clear illustration. Take curve "B0L0" as an example, given the singular values vector as $y$ and the index[13] of each element as $x$, we do the Maximum normalization as $\hat{x} = x/\max(x)$, $\hat{y} = y/\max(y)$, and then plot $\{\hat{x}, \hat{y}\}$.

Figure 15: The singular value of the scale $\mathbf{\Gamma}$ (left) and bias $\mathbf{B}$ (right) within each Conv layer on Flowers.

Based on the discovered low-rank property, we test the compressibility of the learned parameters by truncating/zero-out small singular values. The results in Figure 4(b) are obtained by only keeping $30\% \sim 100\%$ matrix energy [56] at each block alternatively and evaluating their performance (FID). Take B0 as an example, we keep $30\% \sim 100\%$ matrix energy for all the scale $\mathbf{\Gamma}$ and bias $\mathbf{B}$ matrices within block B0 with the other blocks/layers unchanged and evaluate the FID.

To have a straight forward understanding of the proposed compression method for our GAN memory (see Figure 4(c)), we summarize the specific steps in Algorithm 1, where $r = 0.01$ works well in the experiments. Given $\mathbf{E}_t = \mathbf{U}_t \mathbf{S}_t \mathbf{V}_t^T$ with $\mathbf{S}_t = \text{Diag}(\boldsymbol{s}_t)$, the low-rank regularization for the first task is simply implemented via $\min \|\mathbf{E}_1\|_* = \min \|\boldsymbol{s}_1\|_1$; the low-rank regularization for the later tasks are implemented via $\min \|\mathbf{E}_t\|_* = \min \|\boldsymbol{\eta} \odot \boldsymbol{s}_t\|_1$, where $\boldsymbol{\eta}$ is a vector of same size as $\boldsymbol{s}_t$ with non-negative values and is designed as $\boldsymbol{\eta} = 0.1 + \sigma(\frac{10 \cdot j}{J})$ with $\sigma(\cdot)$ as a sigmoid function, $J$ as the length of $\boldsymbol{s}_t$, $j$ as the index of $[\boldsymbol{s}_t]_j$. This kind of designation forms a prior which imposes weak sparse constraint on part of the elements in $\boldsymbol{s}_t$ and imposes strong constraint on the rest. This prior is found effective in keeping a good performance.

---

**Algorithm 1** GAN memory with further compression (exampled by $\mathbf{\Gamma}_t$ within a Conv layer)

---

**Input:** A sequence of T tasks, $\mathcal{D}_1, \mathcal{D}_2, \cdots, \mathcal{D}_T$, the knowledge base $\mathbf{L} = \emptyset$ and $\mathbf{R} = \emptyset$, a scale $r$ to balance between the low-rank regularization and the generator loss.
**Output:** The knowledge base $\mathbf{L}$ and $\mathbf{R}$, the task-specific coefficients, $\boldsymbol{c}_1, \boldsymbol{c}_2, \cdots, \boldsymbol{c}_T$
 1: **for** $t$ in task 1 to task T **do**
 2:     Train GAN memory on the current task $t$ with the following settings:
        $(i)$ $\mathbf{\Gamma}_t \doteq \mathbf{L}_{t-1}^{\text{KB}} \mathbf{\Lambda}_t \mathbf{R}_{t-1}^{\text{KB}}{}^T + \mathbf{E}_t$ with $\mathbf{\Lambda}_t = \text{Diag}(\boldsymbol{\lambda}_t)$;
        $(ii)$ Generator loss with low-rank regularization $r\|\mathbf{E}_t\|_*$ considered;
 3:     Do SVD on $\mathbf{E}_t$ as $\mathbf{E}_t = \mathbf{U}_t \mathbf{S}_t \mathbf{V}_t^T$ with $\mathbf{S}_t = \text{Diag}(\boldsymbol{s}_t)$
 4:     Zero-out the small singular values to keep $X\%$ matrix energy for $\mathbf{\Gamma}_t$ and obtains $\mathbf{E}_t \approx \hat{\mathbf{U}}_t \hat{\mathbf{S}}_t \hat{\mathbf{V}}_t^T$
 5:     Collect the task-specific coefficients for task t as $\boldsymbol{c}_t = [\boldsymbol{\lambda}_t, \boldsymbol{s}_t]$
 6:     Collect knowledge base $\mathbf{L} = [\mathbf{L}, \hat{\mathbf{U}}_t]$, $\mathbf{V} = [\mathbf{V}, \hat{\mathbf{V}}_t]$;
 7: **end for**

---

## G.2 Experiments on the compressibility of the GAN memory

From Figure 4(b), we observe that $\mathbf{\Gamma}/\mathbf{B}$ from different blocks/layers have different compressibility: the larger the matrix size for $\mathbf{\Gamma}/\mathbf{B}$ (the closer to the noise), the larger the compression ratio is. Thus, it is proper to keep different percent of matrix energy for different blocks/layers to minimize the decline in final performance. The results shown in Table 1 are obtained by keeping $\{80\%, 80\%, 90\%, 95\%\}$ matrix energy for $\mathbf{\Gamma}/\mathbf{B}$ within B0, B1, B2, B3 respectively. The compression techniques are not considered for the other blocks/layers (*i.e.,* B4, B5, B6 and FC) because these layers have a relatively small amount of parameters.

Figure 16: The percentage of newly added bases for scale $\mathbf{\Gamma}$(left) and bias $\mathbf{B}$(right) after the sequential training on each of the 6 butterfly tasks. Each bar value is a ratio between the number of newly added bases and the maximum rank of $\mathbf{\Gamma}/\mathbf{B}$. The percentages are the ratio between the amount of newly added parameters with and without using compression for each task.

By comparing to a naive implementation with task-specific style parameters (see Section 4.2), Figure 16 shows the the overall (and block/layer wise) compression ratios delivered by the compression techniques as training proceeds. Each bar shows the ratio between the number of newly added bases and the maximum rank of $\mathbf{\Gamma}/\mathbf{B}$. The shown percentages above each group of bars are the ratio between the amount of newly added parameters with and without using compression for each task. It's clear that ($i$) even for task $\mathcal{D}_1$ (with an empty knowledge base), the low-rank property of $\mathbf{\Gamma}/\mathbf{B}$ enables a significant parameter compression; ($ii$) based on the existing knowledge base, much less new parameters are necessary to form a new generation power (*e.g.,* for task $\mathcal{D}_2$, $\mathcal{D}_3$, $\mathcal{D}_4$, and $\mathcal{D}_5$), confirming the reusability/sharability of existing knowledge; and ($iii$) the lower the block (the closer to the noise with a larger matrix size for $\mathbf{\Gamma}/\mathbf{B}$), the larger the compression. All these three factors together lead to the significant overall compression ratios.

In addition, we also plot in Figure 17 the generated images along the task sequence, to show the effectiveness of GAN memory in situations where the tasks in the sequence are related.

Intuitively, when more tasks are learned and the knowledge base is rich enough, no new bases would be necessary for future tasks (despite we still need to save tiny task-specific coefficients). For the proposed compression method, the selection of a proper threshold for keeping matrix energy is very important in balancing between saving parameters and keeping performance (*e.g.,* when the task number becomes large, the negative effect of the selected threshold on FID performance emerges, see Table 1), we leave this as future research direction.

# H    On the robustness of GAN memory to the source model

Our method is deemed considerably robust to the (pretrained) source model, because ($i$) when we modulate a different source GAN model (pretrained on LSUN Bedrooms) to form the target generation on Flowers, the resulting performance (FID=15.0) is comparable to that shown in the paper (FID=14.8 with CelebA as source); similar property about FC→B6 (now B5 due to the architecture change [49]) is also observed, as shown in Figure 18; ($ii$) the experiments of the paper have verified that, with our style modulation process, various target domains consistently benefit from the same source model (with a better performance than fine-tuning), in spite of their perceptual distances to

Figure 17: Realistic replay of GAN memory on a sequence of relatively related tasks: 6 categories of butterfly images.

the source model; $(iii)$ both $(i)$ and $(ii)$ further confirm our insights in Section 4.1 from the main paper, *i.e.,* GANs seem to capture an underlying universal structure to images (shape within kernels), which may manifest as different content/semantics when modulated with different "styles;" from another perspective, $(i)$ and $(ii)$ also imply that universal structure may be widely carried in various "well-behaved" source models. Therefore, we believe our method and the properties in Section 4.2 could generalize well on different source models.

Figure 18: When selecting a GAN pretrained on LSUN Bedroom, similar properties discovered in Figure 2(c) of the main paper are observed: modulations in different blocks have different strength/focus over the generation.

## Footnotes

[12]The last task have its real data available, thus there is no need to learn to replay

[13]The index here begins from 0.