[Reviews · NeurIPS 2020]

Review 1

Summary and Contributions: Update: Having seen all the other reviews and the author responses, I continue to support acceptance of this paper. This paper presents a replay-based approach for lifelong/incremental learning. In particular, it leverages a generative model to produce samples corresponding to previous tasks/data when adapting the learning framework to a new task/data. A GAN model is used as the framework for generating data, which in itself is "modulated" as and when new data is seen, inspired by style-transfer techniques. Two techniques, namely FiLM and AdaFM, are used to achieve this modulation step, which allow a base GAN model to adapt. This approach is evaluated on the image classification task, and compared with related work.

Strengths: * The paper presents an interesting idea for incremental learning. The idea of adapting the generative model also incrementally is worth noting. * The approach is reasonaly well explained, barring a few details that are missing in the main paper (see below). * The approach is evaluated at different stages, including the data generation steps, with several ablation studies. * Source code (which I have not had a chance to test) is included along with the submission.

Weaknesses: * At several points the paper refers to a well-behaved GAN. This remains vague. How does the initial GAN model trained on CelebA become a well-behaved GAN? What is the impact of learning the initial GAN model on other datasets? How sensitive does the adaptation process become in this case? There are some of the questions that need clarification in the paper. * Section 4.2 defines properties of GAN memory. I do wonder how dependent some of these properties are on the initial "well-behaved" GAN model? Especially, how generic are the observations made about FC to B6 blocks. On a related note, it may be useful to show these blocks in a schematic figure, at least in the supplementary material. * An important comparison in the context of incremental learning is showing the upper bound, i.e., the performance when the model is trained on all the tasks/data jointly. This needs to be in the paper.

Correctness: Some of the claims need clarity (see above).

Clarity: More or less, but some technical details are missing in the main paper (see additional comments below).

Relation to Prior Work: Yes

Reproducibility: No

Additional Feedback: * Important technical details do need to be in the main paper. For example, the approach and the protocol for incremental learning with replay should be in the main paper, at least briefly. The current level of detail for this is insufficient (some of this is in the supplementary material, but it is really needed in the main paper). * Another aspect I was wondering about is the impact related tasks/data have on the generative model. For example, if the sequential tasks were based on data from cat, dog, horse, and cow classes, how would the generated samples be? Is the model robust to such data? * On a philosophical level, should the transfer process be treated as content transfer instead of style? I would associate style with things like a painting, or filters. What is changing here, atleast in the examples (Celeb -> flowers -> buildings -> cat) is the content of the image. * Other comments: - The paper needs to be improved in terms of writing. There are also several typos and grammatical errors. On a related note, the abstract of the paper is very cryptic. - The availability of source code should be mentioned in the paper.


Review 2

Summary and Contributions: This paper adapts feature modulation to continually learn new image generation tasks by only learning new style modulation parameters per task. The backbone GAN remains fixed, which results in no forgetting. The authors adapt FiLM and AdaFM to the continual learning setting. To save memory in the new parameters, they can be compressed with a low rank decomposition. The method is evaluated for continual image generation and classification in a challenging setting with six tasks.

Strengths: - No forgetting by design, while having high quality in generated images. - The approach is sound, and the evaluation too. - The method is scalable to many tasks and efficient (relatively few parameters per task). - The paper provides interesting insights on how style parameters adapt the image generator to other domains.

Weaknesses: - Heavily depends on a powerful source model, which in turn depends on the source domain. - The capacity grows every new task (in contrast to MerGAN, although it requires an additional copy of the generator). - The technical novelty still remains largely incremental. Using style modulation to adapt GANs to other domains was already proposed in BSA[54] and AdaFM. Similarly, mFiLM and mAdaFM are minor tweaks of FiLM and AdaFM.

Correctness: Yes, they are correct.

Clarity: Yes, easy to follow and understand.

Relation to Prior Work: Yes.

Reproducibility: Yes

Additional Feedback: For better understanding, please include in Table 1 the number of new parameters without compression, and the total number of parameters (including the fixed source model).


Review 3

Summary and Contributions: The paper suggests a way to train GANs on multiple datasets sequentially (lifelong learning) using a modulation trick from style transfer literature. They demonstrate that it is possible to reuse the same frozen GAN to generate very different data like faces, flowers and even brain scans. It opens an opportunity to use such a model for generative replay in lifelong learning scenarios. Because the main model is frozen it demonstrates no forgetting. However, it comes at a cost of added style parameters that can take about 20% of the main parameter count. The authors also suggest a trick to compress them further.

Strengths: It is an interesting work that presents a counter-intuitive discovery that GANs seem to learn very general principles of image decomposition which can be reused to completely different datasets with style transfer techniques. It is especially interesting because usually style transfer can't alter image so much to make a brain scan out of a flower.

Weaknesses: Evaluation can be stronger. While the authors demonstrate that GAN can learn multiple datasets, but there are no experiments measuring how well they do this compared to GANs learning these datasets from scratch. I consider this an important part of any lifelong learning paper because very few would want to use tricky generative replay techniques if they underperform significantly. After all compute is getting cheaper every day. The authors claim that their model can be used on streaming datasets (L191), but don't explain it well. In my opinion streaming dataset explicitly doesn't have task boundaries and may manifest very smooth variation of images. It is difficult to capture it with task-specific parameters. Perhaps it is worth mentioning what scenario they had in mind.

Correctness: I believe the method and the claims are correct.

Clarity: The paper is well-written and easy to read.

Relation to Prior Work: The authors compare their work with MeRGAN, relatively recent memory replay GAN, and demonstrate better performance. Related work can be a bit longer, because it is a well-studied task after all. In general lifelong learning is a problem for classification networks, even more so for generative ones.

Reproducibility: Yes

Additional Feedback: In L272 there is a mention that Figure 5 compares the suggested method with MeRGAN, but the Figure doesn't have MeRGAN samples. I suggest also to find a catchy name for their work. It is much easier to cite it in the text if it is *-GAN or something like that. The exact type of GAN used in this work is mentioned only in supplementary. I consider it quite important info, there is no such thing as "just a GAN" nowadays. POST REBUTTAL: I found the rebuttal satisfying and decided to keep positive rating.


Review 4

Summary and Contributions: This paper proposes to learn a GAN on several dataset sequentially by affine transforming its parameters. The main contributions are: 1) Demonstrate that by affine transforming the parameters of a pre-trained GAN, one can adapt it to new domains. 2) Utilize 1) to learn a GAN on several dataset of different image classes sequentially. 3) Propose a method to compress the parameters in the affine transformation process.

Strengths: The paper is well written and easy to follow. The idea of affine transforming the parameters of a network so that it can be adapted to new domains is interesting. The paper includes detailed analysis and experiments for the affine transformation. The results look promising compared to baselines.

Weaknesses: The term style transfer in the paper is misleading. Style transfer refers to the specific task of transferring the style of an image to another image, existing methods achieve so by normalizing and transforming the intermediate features. However, in this paper, the GAN is adapted to the new domain by affine transformation on its parameters rather than learned features. So the concept of style transfer does not seem fit here. Also, it is not clear the major difference of this paper compared to [95]. The core novelty of this paper lies in equation (2) and (3), which are also explored by [95]. What's the major novelty in method compared to [95]?

Correctness: Yes.

Clarity: Yes.

Relation to Prior Work: Please see the weakness section, that this work may add more discussion of its novelty on methods compared to [95].

Reproducibility: Yes

Additional Feedback: Generally the paper is well written, but some parts could be a bit confusing. For instance, what's the definition of a well-behaved GAN? Also, line 114 says W_hat is used to convolve with input feature maps, but in Eq. (3) it is used to convolve with the Conv filter. The citation of [95] misses publication conference.

[Author Response · NeurIPS 2020]

We thank all reviewers for the constructive comments. Below we first address common concerns and then respond to each reviewer
to address the rest. The paper will be revised correspondingly (*e.g.,* to correct typos/errors, to move important technical details to the
main paper, to revise the abstract and the broader impact carefully to better reveal the delivered information). Code will be released.

**On our novelties over BSA [57] and AdaFM [95].** We consider our main novelties/contributions as (*i*) we reveal that one can
modulate the "style" of a GAN model to form perceptually-distant but realistic targeted generation, which essentially implies GANs
capture a certain universal structure to images (see L139-L141); the realistic generation from that style modulation also reveals
another orthogonal dimensionality for transfer learning, which potentially outperforms/complements the commonly used finetuning
(see L172-L181); (*ii*) we leverage (*i*) to deliver the GAN memory for lifelong learning, which has growing generative power yet with
no forgetting; (*iii*) we generalize our GAN memory with compression techniques, and to *conditional* GAN. In addition to the these
novelties/contributions, and generalizing FiLM/AdaFM to mFiLM/mAdaFM (see Eqs. (4)-(5)), we also empirically reveal/analyze
the role each component of mFiLM/mAdaFM plays (see Sec. 4.2). None of these contributions have been made before or in [57,95]
(see L84-L90 for the differences between our GAN memory and [57,95]).

**On well-behaved source GAN models.** By "well-behaved" we mean the shape within kernels (see L175; shared between source
and target) is well trained. Empirically, this requirement can be *readily satisfied* if the source model (*i*) is pretrained on a (moderately)
large dataset (*e.g.,* CelebA; often a dense dataset is preferred [83]) and (*ii*) it's sufficiently trained and shows relatively high generation
quality. That means many pretrained GAN models can be "well-behaved," including the adopted GP-GAN pretrained on CelebA.
One can of course expect better performance if a better source model (pretrained on a large-scale dense and diverse dataset) is used.

**On the robustness to the source model.** Our method is deemed considerably robust to the (pretrained) source model: (*i*) we
modulated a different source GAN model (pretrained on LSUN Bedrooms) to form the target generation on Flowers; the resulting
performance (FID=15.0) is comparable to that shown in the paper (FID=14.8 with CelebA as source); similar property about FC→B6
(now B5 due to the architecture change [49]) is also observed, as shown in Fig. A; (*ii*) the experiments of the paper have verified
that, with our style modulation process, various target domains consistently benefit from the same source model (with a better
performance than finetuning), in spite of their perceptual distances to the source model; (*iii*) both (*i*) and (*ii*) further confirm our
insights (L139-L141), *i.e.,* GANs seem to capture an underlying universal structure to images (shape within kernels (Line 175)),
which may manifest as different content/semantics when modulated with different "styles;" from another perspective, (*i*) and (*ii*) also
imply that universal structure may be widely carried in various "well-behaved" source models. Therefore, we believe our method and
the properties in Sec. 4.2 could generalize well on different source models. We'll add more demonstrations/discussions to support
our statements.

**On "style".** We consider our mFiLM/mAdaFM as style-transfer techniques, because they are motivated from and mathematically
similar to those techniques, *i.e.,* to manipulate means and standard derivations. But the "style" here (mean/standard-derivation of
kernels) is indeed different from or generalizes over style-transfer literature (see L125-L129); by "style" we mean the style *of a*
*function* (*e.g.,* a generator, a discriminator, and potentially a classifier). For a generator, its "style" may manifest as the content of
generation, which however may not fit a discriminator/classifier. To pay our respect to and distinguish from style-transfer literature,
we used the term "style modulation (of a function)" instead. We'll elaborate more on this; proposals are extremely appreciated.

A. This pretrained model only has FC→B5                                  B                                              C                                              D

**Reviewer #1**: Please see our responses above on common concerns. We'll add the suggested upperbound (labeled as "Joint" in
Fig. B) and move important technical details to the main paper. We've actually considered both diverse and related/similar target
tasks in Secs. 5.1 and 5.3, respectively. In Sec. 5.3, we considered 6 sequential tasks on butterfly images (one category per task; see
L298-L300; Fig. C shows the generated samples). Thus, our method is believed robust to both diverse and related (image) tasks.

**Reviewer #2**: On one hand, our GAN memory has moderate requirements for and is considerably robust to the source model, thanks
to its style modulation process (see our responses above); on the other hand, many powerful pretrained GANs have been released
[95] and the valuable information therein (often benefiting downstream tasks greatly [83,95]) is one motivation for our method.
In lifelong-learning settings, a growing model capacity might be necessary; when compared with MeRGAN (see L71-L78), our
GAN memory works much better (see Fig. 6) with good properties (see Footnote 4); further considering its compression potential,
we believe our GAN memory may serve as a practical/realistic generative replay for lifelong-learning problems. Please see our
responses "On our novelties over BSA [57] and AdaFM [95]." We'll revise Table 1 following your suggestions.

**Reviewer #6**: Usually, our method is expected to outperform learning from scratch, because (*i*) our method shows better training
efficiency and performance than finetuning (see Fig. 1(b)) and (*ii*) by referring to [83] and the transfer learning literature, finetuning
from pretrained models often outperforms scratch on efficiency and performance. We empirically verified that by running scratch
on Flowers (see Fig. D). To learn a GAN on (rigorous) streaming datasets (one image per task) is extremely challenging. Our
streaming setting is actually a practical work-around, *e.g.,* by leveraging a physical memory buffer to form a stream of datasets
with clear task boundaries. We'll add discussions and remove misleading terms. We'll enrich discussion of related work with more
citations/discussions. The samples from MeRGAN are shown in the bottom two rows of Fig. 5(right); please zoom in for details.

**Reviewer #9**: Please see our responses "On our novelties over BSA [57] and AdaFM [95]" and "On well-behaved source GAN
models." The question about $\hat{\mathbf{W}}$ is a little confusing; the notation $\odot$ denoting the Hadamard product might be misunderstood as a
convolution; Eq. (3) shows how $\hat{\mathbf{W}}$ is calculated, after which $\hat{\mathbf{W}}$ is then convolved with input feature maps.

[Meta-Review · NeurIPS 2020]

The paper presents an GAN-based method to learn from multiple datasets sequentially. To adapt to multiple datasets, the authors propose modulation tricks from style transfer, e.g., FiLM and AdaFM, which allows for generating different types of data without forgetting. The reviewer all agree on acceptance for the following reasons: interesting idea for incremental learning, addressing the catastrophic forgetting issue with a reasonable approach, strong experimental results, etc. This AC agrees to accept the paper.